# Deciphering Machine Learning Decisions to Distinguish between Posterior Fossa Tumor Types Using MRI Features: What Do the Data Tell Us?

**DOI:** 10.3390/cancers15164015

**Published:** 2023-08-08

**Authors:** Toygar Tanyel, Chandran Nadarajan, Nguyen Minh Duc, Bilgin Keserci

**Affiliations:** 1Department of Computer Engineering, Yildiz Technical University, Istanbul 34349, Türkiye; toygar.tanyel@std.yildiz.edu.tr; 2Department of Radiology, Gleneagles Hospital Kota Kinabalu, Kota Kinabalu 88100, Sabah, Malaysia; nadarajan.chandran@gleneaglesdr.my; 3Department of Radiology, Pham Ngoc Thach University of Medicine, Ho Chi Minh City 700000, Vietnam; bsnguyenminhduc@pnt.edu.vn; 4Department of Biomedical Engineering, Yildiz Technical University, Istanbul 34349, Türkiye

**Keywords:** posterior fossa pediatric brain tumors, magnetic resonance imaging, machine learning, exploratory data analysis, kernel density estimation

## Abstract

**Simple Summary:**

This paper focuses on interpreting machine learning (ML) models’ decisions in medical diagnoses, specifically for four types of posterior fossa tumors in pediatric patients. The proposed methodology involves using kernel density estimations with Gaussian distributions to analyze individual MRI features, assess their relationships, and comprehensively study ML model behavior. The study demonstrates that employing a simplified approach in the absence of large datasets can lead to more pronounced and explainable outcomes. Furthermore, the pre-analysis results consistently align with the outputs of ML models and existing clinical findings. By bridging the knowledge gap between ML and medical outcomes, this research contributes to a better understanding of ML-based diagnoses for pediatric brain tumors.

**Abstract:**

Machine learning (ML) models have become capable of making critical decisions on our behalf. Nevertheless, due to complexity of these models, interpreting their decisions can be challenging, and humans cannot always control them. This paper provides explanations of decisions made by ML models in diagnosing four types of posterior fossa tumors: medulloblastoma, ependymoma, pilocytic astrocytoma, and brainstem glioma. The proposed methodology involves data analysis using kernel density estimations with Gaussian distributions to examine individual MRI features, conducting an analysis on the relationships between these features, and performing a comprehensive analysis of ML model behavior. This approach offers a simple yet informative and reliable means of identifying and validating distinguishable MRI features for the diagnosis of pediatric brain tumors. By presenting a comprehensive analysis of the responses of the four pediatric tumor types to each other and to ML models in a single source, this study aims to bridge the knowledge gap in the existing literature concerning the relationship between ML and medical outcomes. The results highlight that employing a simplistic approach in the absence of very large datasets leads to significantly more pronounced and explainable outcomes, as expected. Additionally, the study also demonstrates that the pre-analysis results consistently align with the outputs of the ML models and the clinical findings reported in the existing literature.

## 1. Introduction

Brain tumors are the most prevalent type of childhood cancer, comprising over a quarter of all cases. Among these tumors, 60–70% arise in the posterior fossa (PF), with medulloblastoma (MB), ependymoma (EP), pilocytic astrocytoma (PA), and brainstem glioma (BG) being the most common types in children. These tumors can negatively impact mental and physical development.

Clinical information from radiological interpretations and the histopathological features of tumors plays a crucial role in diagnosing, prognosticating, and treating PF tumors in children. Histopathological evaluation, which is necessary for the initial diagnosis, helps to evaluate patient prognosis and direct clinical and therapeutic management. It remains the gold standard in differentiating PF tumors [1,2]. Although biopsies of different PF brain tumors can reveal distinct visual characteristics, they carry significant risks of morbidity and mortality, in addition to being expensive. Recent progress in characterizing tumor subtypes based on cross-sectional diagnostic imaging indicates that it can help to predict differential survival and responses to treatment. This development is particularly promising for future treatment stratification in PF tumors. Hence, developing a novel non-invasive diagnostic tool is essential in classifying tumors based on type and grade and aiding in planning treatment.

Magnetic resonance imaging (MRI) is currently the most preferred non-invasive method. It offers high intrinsic soft-tissue contrast without the risk of ionizing radiation. Conventional MRI protocols, including T1-weighted (T1W), T2-weighted (T2W), and fluid-attenuated inversion recovery (FLAIR) MRI sequences, have shown promising results in differentiating types of PF tumors in children [3,4,5,6,7,8,9,10,11,12,13,14,15,16,17,18,19,20,21]. Additionally, diffusion-weighted imaging (DWI) with apparent diffusion coefficient (ADC) maps allows the assessment of physiological features to discriminate between low- and high-grade tumors and their different subtypes [22,23,24,25,26,27,28,29,30,31,32,33,34,35,36,37].

While numerous advancements have been made, the diagnosis and prognosis of specific tumor matches still present significant challenges due to the voxel-wise overlap [23,27,38]. The classification process necessitates the inclusion of a tumor’s molecular profile as a critical variable to predict the diverse biological behaviors of entities that exhibit histological similarities or even indistinguishability [2]. An extensive exploration of tumor classifications has been conducted using MRI in the literature. Nevertheless, accurately distinguishing between these tumor types remains an active area of research [20,39,40,41,42]. The differentiation between MB and EP is of the utmost importance, considering the distinct treatment planning required for each, underscoring the significance of their accurate diagnosis in numerous cases.

Artificial intelligence (AI) applications in pediatric brain tumor research are currently not well documented when compared to the available literature for adults. Challenges arise due to the unique pathology of pediatric cases and limitations in available data, which hinder the development of AI applications specifically tailored to children [43]. While there is growing interest in utilizing AI for pediatric brain tumor classification [44,45,46,47,48,49,50,51,52,53,54,55], the integration of AI into clinical workflows encounters significant obstacles beyond mere classification. One major challenge is the limited interpretability of many AI methods; creating a “black-box” model raises concerns among clinicians and patients. To address this issue, our research aims to enhance the interpretability of ML models’ outcomes, which are frequently either blindly accepted or disregarded due to their black-box nature. To the best of our knowledge, there is a lack of literature specifically focusing on the issue of reasoning and explainability [56].

This study had two main objectives, aiming to bridge the gaps between ML outcomes and medical knowledge. Firstly, it sought to investigate the significance of clinical MRI features in classifying pediatric PF tumors (MB, EP, PA, and BG) through exploratory data analysis (EDA). Secondly, it aimed to offer explanations for the ML outcomes by leveraging the insights gained from the data exploration (Figure 1).

## 2. Materials and Methods

### 2.1. Ethics Statement and Patient Characteristics

This prospective study (Ref: 632 QÐ-NÐ2 dated 12 May 2019) was conducted in both Radiology and Neurosurgery departments and approved by the Institutional Review Board in accordance with the 1964 Helsinki declaration. Prior to the MRI procedure, written informed consent was obtained from the authorized guardians of the patients. The study included 112 pediatric patients diagnosed with PF tumors, including 42 with MB, 11 with EP, 25 with PA, and 34 with BG. All BG patients were confirmed based on full agreement between neuroradiologists and neurosurgeons, while the remaining MB, EP, and PA patients underwent either surgery or biopsy for histopathological confirmation.

The demographics of the patient population were analyzed to gain insights into their age, gender, and weight distributions. The age statistics revealed a mean age of 6.55 years, with a median age of 6.0 years. The age range varied from a minimum of 0.6 years to a maximum of 15.0 years, reflecting the diversity within our cohort. Regarding gender, we observed a greater representation of males, with a count of 68, compared to females, with a count of 44. The mean weight was calculated to be 22.54 kg, with a median weight of 20.5 kg. The range of weights varied from a minimum of 3 kg to a maximum of 48 kg.

In-depth patient demographics can be found in the accompanying Table 1, which provides a comprehensive overview of the study population. Table 1 includes detailed information on gender, age, and weight for the patients.

### 2.2. Data Acquisition and Assessment of MRI Features

The MRI protocol was performed in the supine position using a 1.5 Tesla MRI scanner (Philips, Best, The Netherlands) and included T1W, T2W, FLAIR, DWI (b values: 0 and 1000) with ADC, and T1 contrast-enhanced (T1CE) sequences with macrocyclic gadolinium-based contrast enhancement (0.1 mL/kg Gadovist, Bayer, Germany or 0.2 mL/kg Dotarem, Guerbet, France).

MR images of all patients were imported into the Medical Imaging Interaction Toolkit, developed by the German Cancer Research Center’s Division of Medical Image Computing in Heidelberg, Germany. The radiologists precisely identified the slice in which the largest diameter of the PF tumor was present. For each patient, ROIs corresponding to the posterior fossa tumors and normal-appearing parenchyma were manually delineated on the T1W, T2W, FLAIR, DWI, and ADC images. These delineations were based on the consensus reached by two expert radiologists with over 10 years of experience in interpreting neuro MR images. An example of ROI delineation on a T2W MRI is provided in Figure 2. For additional ROI delineations of other sequences, please refer to Appendix A.

The following MRI features were evaluated: signal intensities (SIs) of T2, T1, FLAIR, T1CE, DWI, and ADC. The ratio of each MRI feature was calculated as the quotient of the tumor’s SI and the SI of the normal-appearing parenchyma ($\mathrm{Ratio}=\frac{\mathrm{Tumor}\;\mathrm{Feature}}{\mathrm{Parenchyma}\;\mathrm{Feature}}$). Additionally, ADC values were quantified for both the PF tumor and parenchyma regions using the MR Diffusion tool available in the Philips Intellispace Portal, version 11 (Philips, Best, The Netherlands).

### 2.3. Exploratory Data Analysis

The quality of a dataset has a direct impact on the effectiveness of the trained model. Therefore, EDA plays a crucial role in understanding the data by revealing its inherent structure, identifying anomalies and outliers, extracting significant features, and facilitating the appropriate ML models to establish correlations between MRI feature characteristics and the various types of pediatric PF tumors.

In this study, we performed an exploratory analysis using kernel density estimations (KDE) with Gaussian distributions, focusing on the MRI features. The proposed analysis consisted of three parts: standardization, data analysis involving visualization of the distributions of each MRI feature, as well as exploring relationships between different features, and analyzing the ML models’ outcomes through the extracted knowledge from EDA. All figures were generated using the Matplotlib package (version 3.5.2) in Python.

### 2.4. Standardization

The patient dataset underwent a standardization process known as Z-score normalization. This process was carried out using the Python programming language, specifically Python version 3.9.13, along with the Scikit-Learn library version 1.0.2.

To perform the standardization, the StandardScaler function from Scikit-Learn was utilized. This function ensured that numerical attributes within the patient dataset were transformed into a standardized format. It achieved this by subtracting the mean and scaling the values to have a unit variance.

The StandardScaler function normalizes each feature individually, meaning that each column/feature/variable in the input matrix *X* will have a mean (μ) of 0 and a standard deviation (σ) of 1. The normalization is accomplished using the formula $z=\frac{x_{i}-\mu }{\sigma }$, where $x_{i}$ represents the value of a specific feature for a patient.

### 2.5. Pairwise Feature Analysis

The pairplot function in the Seaborn Python package (version 0.11.2) enables the visualization of the pairwise relationships between variables in a dataset. Numerical variables are split into a single row on the y-axis and a single column on the x-axis by default. The position of one variable on the vertical or horizontal axis indicates its correlation with another variable in the same row of data. The relationship between the MRI features was further examined through Pearson’s correlation coefficients, calculated using the default corr() function in the pandas dataframe.

### 2.6. Revealing Distribution Differences of Patients between Tumor Types

To effectively illustrate the distinctions among the four PF tumor types, we utilized the kdeplot and pairplot functions from Seaborn as necessary. Additionally, we assigned a hue parameter to represent the tumor type, thereby facilitating a semantic mapping. This assignment transforms the default marginal plot into a layered KDE, which helps to address the challenge of reconstructing the density function *f* using an independent, identically distributed (iid) sample $x_{1},x_{2},...,x_{n}$ from the respective probability distribution.

The generalized estimate used in plotting can be expressed as follows:(1)$$\hat{f}\left(x\right)=\frac{1}{nh^{d}}\sum_{i=1}^{n}k\left(\frac{x-x_{i}}{h}\right),\;$$
where *h* is a bandwidth parameter, and the kernel is commonly a Gaussian,
(2)$$k\left(z\right)=\frac{1}{\sqrt{2\pi }}\exp\left(-\frac{1}{2}z^{2}\right).\;$$

### 2.7. Machine Learning

We employed eight ML models, including support vector machine (SVM), linear support vector machine (LSVM), logistic regression (LR), a random forest classifier (RF), a decision tree classifier (DT), a gradient boosting classifier (GBM), a catboost classifier (CB), and an extreme gradient boosting classifier (XGB), to assess the consistency of our interpretations of the raw data with the outcomes. CB and XGB were obtained from their respective libraries (CatBoost version 1.1.1, XGBoost version 1.5.1), while other models were obtained from the Scikit-Learn library.

To ensure methodological consistency, we utilized the default versions of all ML models, as our primary objective was not to maximize the classification scores. It is important to acknowledge that tuning the model parameters could potentially lead to improved results. However, considering the limited data size, the presence of rare tumor types, and the absence of an external dataset from another hospital, such parameter adjustments carried a significant risk of overfitting on our data. In order to mitigate this risk and uphold the credibility of our findings, we chose to adhere to the default model configurations throughout the analysis. This decision safeguarded the integrity of our research and ensured the validity of our conclusions.

Tree-based models, such as RF, DT, GBM, CB, and XGB, are commonly utilized in ML. However, they can be prone to overfitting when the trees are deep and have a large number of features. To address this issue, RF, which is a bagging model, generates a set of decision trees by training on different data samples or subsets of features. XGB, on the other hand, is a sequential model that adopts a different approach to building decision trees. To ensure that our models did not have a bias towards certain features and generalized well, we conducted an analysis of the prioritization and proportional distribution of features used by the RF and XGB models during prediction. This analysis strengthened our explanations of the models’ high performance and accuracy in predicting outcomes.

To ensure the reliability of our ML models, particularly with a small dataset, we conducted five runs using stratified random sampling based on tumor type with 55% train and 45% test patients. We used random states to obtain samplings and preserve the train/test distributions for the reproducibility of the experiment. Ultimately, we calculated the averaged outcomes with their standard deviations.

The accuracy metric is not employed in the presentation of our results due to significant class imbalance in our dataset. Utilizing the accuracy metric could have led to misleading results. Instead, we relied on the precision, recall, and F1 score, computed based on the number of true positives (TP), true negatives (TN), false positives (FP), and false negatives (FN), as fundamental evaluation metrics to assess the performance of our classifiers in both binary and multiclass classification tasks. Precision gauges the proportion of correctly predicted positive instances among all positive predictions, highlighting the accuracy of positive classifications. Conversely, recall assesses the proportion of positive instances that were correctly identified by the classifier, emphasizing the completeness of the positive predictions.

While high precision and high recall are typically desirable, we were aware of the potential trade-off between these two metrics in certain scenarios. To gain a comprehensive understanding of our classifier’s effectiveness, we utilized the F1 score, which harmoniously considers both precision and recall.

To ensure a precise interpretation of the ML results, we chose not to equalize the labels. Instead, we utilized the macro precision, macro recall, and macro F1 score metrics to ensure that all labels contributed equally to the results. This approach allowed us to assess the classifier’s performance while considering the impact of varying patient counts across different labels.

The validation metrics used in ML are as follows:(3)$$\mathrm{Macro}\;\mathrm{Precision}=\frac{1}{n}\sum_{i=1}^{n}\frac{{\mathrm{TP}}_{i}}{{\mathrm{TP}}_{i}+{\mathrm{FP}}_{i}}$$
(4)$$\mathrm{Macro}\;\mathrm{Recall}=\frac{1}{n}\sum_{i=1}^{n}\frac{{\mathrm{TP}}_{i}}{{\mathrm{TP}}_{i}+{\mathrm{FN}}_{i}}$$
(5)$$\mathrm{Macro}\;F1\;\mathrm{Score}=\frac{1}{n}\sum_{i=1}^{n}\frac{2\times {\mathrm{TP}}_{i}}{2\times {\mathrm{TP}}_{i}+{\mathrm{FP}}_{i}+{\mathrm{FN}}_{i}}$$
where *n* represents the total number of classes.

### 2.8. Statistical Analysis

Statistical analysis was performed using the SPSS software (version 25.0, 64-bit edition, IBM Corp., Armonk, NY, USA). A two-sided *p*-value of <0.05 was considered statistically significant. The statistical summary of the variability of ML outcomes is presented in the mean ± standard deviation format.

### 2.9. Hardware Requirements for Machine Learning

Designing an ML pipeline with the current number of patients and their tabular data does not require significant computational power. The entire machine learning system was developed utilizing a system equipped with an Apple M1 chip CPU and a memory capacity of 16 GB, namely the Hynix LPDDR4.

## 3. Results

### 3.1. Single MRI Feature Analysis

Our feature analysis, which utilized KDE on the standardized distributions presented in Figure 3a–f, yielded several valuable insights.

T2_Tumor features possess distributions that are expected to differentiate PA from EP and MB but cannot differentiate between MB and EP or PA and BG (Figure 3a). Moreover, the T2_Ratio might aid in distinguishing between MB and EP, as well as PA and BG.

The distributions of FLAIR_Tumor and FLAIR_Ratio generate notably different distributions (Figure 3b), even though the Ratio feature is mathematically dependent on the Tumor feature. The FLAIR distributions might be effective in distinguishing between MB and EP, as demonstrated by FLAIR_Tumor, which exhibits a broad EP and a narrow MB distribution. Furthermore, the FLAIR_Ratio exhibits two distinct and narrow Gaussian distributions, which also might aid in distinguishing between MB and EP. In contrast, the other scenarios do not present any discriminative characteristics.

The DWI characteristics (Figure 3c) demonstrate distributions that allow differentiation between MB and PA. Additionally, although to a lesser extent, discrete distributions can be observed in the differentiation between MB and BG, as well as between EP and PA. On the other hand, despite their high distinctive distributions overall, DWI_Ratio features are not expected to be effective in distinguishing between PA and BG due to significant overlap.

ADC (Figure 3d) demonstrates separate distributions in distinguishing each tumor pair, with the highest distinction observed between MB and PA and the least between PA and BG. When considering tumors as a whole rather than in pairs, ADC and DWI present the most distinct distributions for all tumor types. ADC shows highly distinct distributions for each tumor, with DWI following closely behind.

The T1 features, as shown in Figure 3e, do not demonstrate any distinctive distributions that can effectively differentiate between different tumor scenarios. However, the T1_Ratio appears to be a critical factor in distinguishing PA from other types of tumors. In addition, T1CE presents important distinct distributions for all other tumor matches with BG, as depicted in Figure 3f.

### 3.2. Pairwise Analysis of MRI Features

The scatter correlation plots (Appendix A) and Pearson’s correlation coefficients (Figure 4) illustrate varying degrees of correlation between the MRI features and tumor types. Notably, MB exhibits clustered shapes, while PA appears scattered in most cases, and BG and EP show dispersed and uncertain distributions. Outlier patients with correlated features were identified, and some features exhibited no correlations with the tumor types.

The results of the Pearson’s correlation analysis indicated that the T2 and ADC features, with complex distributions compared to other features, exhibited significant positive correlations, particularly T2_Tumor and ADC_Tumor (r = 0.87, *p* < 0.0001), T2_Tumor and ADC_Ratio (r = 0.85, *p* < 0.0001), T2_Ratio and ADC_Tumor (r = 0.78, *p* < 0.0001), and T2_Ratio and ADC_Ratio (r = 0.79, *p* < 0.0001). Conversely, significant negative correlations were observed between the T2 and DWI features, as well as between the DWI and ADC features, namely T2_Tumor and DWI_Tumor (r = −0.46, *p* < 0.0001), T2_Tumor and DWI_Ratio (r = −0.52, *p* < 0.0001), T2_Ratio and DWI_Tumor (r = −0.51, *p* < 0.0001), T2_Ratio and DWI_Ratio (r = −0.44, *p* < 0.0001), ADC_Tumor and DWI_Tumor (r = −0.68, *p* < 0.0001), ADC_Tumor and DWI_Ratio (r = −0.79 *p* < 0.0001), ADC_Ratio and DWI_Tumor (r = −0.66, *p* < 0.0001), and ADC_Ratio and DWI_Ratio (r = −0.78, *p* < 0.0001).

FLAIR_Tumor did not demonstrate a significant correlation with any other features (T2_Tumor (r = 0.08, *p* = 0.39), T1_Tumor (r = 0.02, *p* = 0.84), T1CE_Tumor (r = −0.09, *p* = 0.34), DWI_Tumor (r = 0.02, *p* = 0.85), and ADC_Tumor (r = 0.06, *p* = 0.52)), while FLAIR_Ratio could exhibit correlations in logarithmic or reduced dimensions (T1_Ratio (r = 0.25, *p* = 0.008). Similar patterns to FLAIR were observed for T1_Tumor (T2_Tumor (r = −0.16, *p* = 0.08), T1CE_Tumor (r = 0.07, *p* = 0.45), DWI_Tumor (r = 0.03, *p* = 0.76), and ADC_Tumor (r = −0.12, *p* = 0.19)), and T1_Ratio (T2_Tumor (r = −0.47, *p* < 0.0001), T2_Ratio (r = −0.53, *p* < 0.0001), T1CE_Tumor (r = −0.19, *p* = 0.045), T1CE_Ratio (r = −0.21, *p* = 0.03), ADC_Tumor (r = 0.41, *p* < 0.0001) and ADC_Ratio (r = 0.40, *p* < 0.0001)), emphasizing the importance of using a ratio computed with reference to parenchyma. In contrast, T1CE_Tumor and T1CE_Ratio showed dispersed distributions, with non-linear patterns that could be observed for certain tumor types.

### 3.3. Findings from Machine Learning

The ML procedure involved analyzing feature importance scores, the test scores of eight ML models (Appendix A), and confusion matrices to assess the accuracy and reliability of the results. We trained the models on all possible tumor pairs to identify unique features for each case, and the most favorable outcomes are summarized in Table 2. Additionally, we conducted a comprehensive analysis of the feature importance scores for all four tumor types, providing further insights into their distinguishing characteristics.

We focused on the RF and XGB models since RF delivered the best scores for the classification of all tumors, while XGB possesses a distinct tree structure compared to RF, allowing us to explore and compare the variations in the ML models’ outcomes. Although various ML models could have been employed for this analysis, we specifically chose XGB and RF to illustrate how the methods’ structures differ in generating importance scores and to present a clear and concise analysis.

Notably, as shown in Figure 5a, the FLAIR_Ratio was identified as the most discriminating feature in distinguishing between MB and EP in both the RF and XGB models, followed by the ADC_Ratio. However, the two models relied on different features for decision-making. Therefore, relying solely on the analysis presented in Figure 5 may not be sufficient for model comparison, as they prioritize different features. The performance evaluation of both models showed that the RF model, which prioritized diffusion features (4 out of top 5, 65.38%), demonstrated greater accuracy in feature selection compared to the XGB model (2 out of top 5, 60.08%). Therefore, the features highlighted by the RF model should be considered more significant in distinguishing between MB and EP.

In differentiating between MB and PA, the RF model showed a more dispersed reliance on various features, whereas the XGB model heavily relied on ADC_Tumor (Figure 5b). The study suggests that DWI features play an important role in distinguishing between these tumors, with T2 features being crucial in the decision-making process of the RF model, leading to a higher F1 score (RF: 93.81%, XGB: 90.15%).

To differentiate between MB and BG, our results indicated that the XGB model heavily relied on the ADC_Tumor feature (∼80%), while the RF model utilized a more diverse set of features, such as ADC, DWI, T1CE, and T2 (Figure 5c). Surprisingly, both models performed equally well in this scenario, with an F1 score of 93.62%. Therefore, our results suggest that the ADC_Tumor feature is crucial in distinguishing MB from BG.

In differentiating EP from BG, the ML models were mainly impacted by the ADC_Tumor feature, with a supportive influence of the T1CE_Ratio and T1CE_Tumor features (Figure 5d). The RF model was also influenced by T2_Tumor, which might have led to a slight decrease in the overall F1 score (RF: 69.96%, XGB: 73.34%). As there was significant feature overlap between EP and BG, the performance scores for this classification task were lower compared to other tumor pairs, except MB and EP.

In distinguishing EP from PA, T2 features provided significant discriminative power (Figure 5e). However, the ADC_Tumor and ADC_Ratio features were found to be the leading contributors to the F1 score of 92.18% for the RF model, while the XGB model achieved a score of 81.48% due to its strong dependency on T2.

To differentiate PA from BG, both ML models heavily relied on T1CE features in their decision-making processes, with the T2_Ratio also providing discriminative power (Figure 5f). The XGB model outperformed the RF model, achieving a higher F1 score of 89.01% compared to 87.25%.

Additionally, the LR model achieved the highest F1 score in the MB-EP case and emerged as the dominant model in the MB-BG, EP-PA, and EP-BG cases. Furthermore, the LSVM outperformed other models in distinguishing between MB and PA. For the PA-BG case, the CB model attained high scores.

The analysis revealed varying degrees of feature importance in differentiating between the four tumor types in the MB, EP, BG, and PA classification task. Among the models, the RF model achieved the highest F1 score of 71.74%, outperforming the other models. Figure 6c illustrates the significant role of ADC features in overall differentiation, followed by the T1CE and T2 features. The DWI and FLAIR features also contributed to the discriminative power, albeit to a lesser extent.

We also identified the most challenging discrimination task, which involved distinguishing between MB and EP (Figure 6a), and the easiest discrimination task, which involved distinguishing between MB and PA (Figure 6b). In the challenging classification problem of MB and EP, the Gaussian distributions of the best distinguishing features were found to overlap significantly, while, in the easiest one, MB and PA, the distributions of the best features did not overlap at all. Conversely, the least important features overlapped completely in every scenario.

The impact of stratified random sampling on the feature selection and performance of the RF model was examined and the findings are as given below (Appendix A).

In State 1, the model misclassified nine patients, including two with a BG tumor, three with an EP tumor, one with an MB tumor, and three with a PA tumor. The most informative features for this classification were ADC_Tumor, ADC_Ratio, and T1CE_Ratio.In State 2, the model performed slightly better in predicting BG and could distinguish all MB from other types. However, it misclassified two more PA patients, using ADC_Ratio, ADC_Tumor, and T2_Ratio as the most significant features.In State 3, the model could distinguish almost all PA test patients except one. However, it missed one BG, which was previously predicted as EP in State 2.In State 4, the model was unable to differentiate three BG, three EP, and three PA test patients from other types.In State 5, the model attributed the highest importance to the T1CE_Tumor feature, which led to the misclassification of all EP patients and four BG patients.

## 4. Discussion

Pediatric brain tumors pose a significant clinical challenge due to the substantial degree of spatial heterogeneity in tumor characteristics. Tumors such as those arising from the posterior fossa have a significant imaging feature overlap, leading to difficulty in differentiation, even among experts. The need to differentiate is important due to the different treatment options available for each of them. Thus, precise diagnosis and treatment are crucial in improving outcomes and enhancing quality of life.

Despite the significant advancements observed in AI and medical imaging, the dependability and accuracy of these approaches are profoundly influenced by the quality of the data, meticulous system design, and the comprehensive dissemination of transparent results. Therefore, we conducted a comprehensive and systematic analysis focusing on four distinct tumor types in pediatric brain tumor research. We employed an approach that integrated EDA to interpret ML outcomes, while the ML models provided additional insights into the underlying patterns and relationships among the MRI features.

This study was motivated by the idea that the feature distribution obtained from KDE can provide reliable estimates of ML results prior to the actual model training. The estimation provides insights into which features are the most effective and which features contribute negligibly to the ML models’ decisions. To test this hypothesis, we conducted several pre-training analyses without relying on prior clinical knowledge and analyzed the feature distribution plots. In the present research, a thorough investigation of the diverse characteristics of pediatric PF tumor types was carried out through the utilization of Gaussian distributions, which can be observed in Figure 3. Through this analysis, a number of predictions have been drawn, indicating that certain features are likely to be highly effective in distinguishing particular tumor types, while others are deemed to have a limited impact on classification.

The single-feature analysis using Gaussian distributions, shown in Figure 3, revealed that some MRI features are effective in distinguishing specific pediatric PF tumor types, while others have minimal contributions towards classification. ADC and DWI features are the most effective in differentiating between tumor types, with clear differences in the distributions of these features for different tumors, whereas T1 and T1CE features are less effective in distinguishing between tumor types, although there are some differences in the distributions for different tumors. Moreover, T2 and FLAIR features show some differences in the distributions for different tumors, but these are less pronounced than for ADC and DWI.

Our analyses in the single-feature section are consistent with both the clinical and ML results in almost every instance, and we provide corresponding references in this section to validate our findings. Specifically, our analysis indicates that the T2_Tumor feature can effectively differentiate PA from EP and MB (Figure 5b–d), but cannot differentiate between MB and EP or PA and BG (Figure 5a–f). Remarkably, the incorporation of the handcrafted feature T2_Ratio further enhances the effectiveness of T2 for tumor classification (Figure 5a,b,e,f). This is particularly evident in the differentiation between MB and EP, as well as PA and BG tumors (Figure 5a–f). Our findings also shed light on the potential of FLAIR features in distinguishing between different tumor types (Figure 3b and Figure 5). The distributions of FLAIR_Tumor and FLAIR_Ratio exhibit notable differences, despite the lack of distributional disparities for parenchyma, which serves as a reference point. Specifically, FLAIR_Tumor shows a broad distribution for EP and a narrow distribution for MB, while FLAIR_Ratio displays two distinct and narrow Gaussian distributions. The ML results highlight that FLAIR features are useful in distinguishing between MB and EP tumors (Figure 5a), although the discriminative characteristics are not evident in the rest of the scenarios (Figure 5b–f). These findings are consistent with those of previous studies [10,11,12,15,17,19,20].

The results of our study also indicate that the DWI characteristics display distinct distributions that enable the differentiation of MB and PA (Figure 3c and Figure 5b). While the distributions are less clear, there are still noticeable differences in the DWI characteristics when distinguishing between MB and BG, as well as between EP and PA (Figure 3c and Figure 5c,e). However, we found that the DWI_Ratio features, despite having highly distinctive distributions overall, were not likely to be useful in distinguishing between PA and BG due to their significant overlap (Figure 3c and Figure 5f). Moreover, our findings revealed that the ADC had distinct distributions for each tumor pair, with the most noticeable distinction between MB and PA and the least between PA and BG (Figure 3d and Figure 5a–f). When considering the distributions of all tumors collectively, rather than in pairs, ADC exhibited the most prominent differences among all tumor types (Figure 6c). The difference in diffusivity between various types of PF tumors is due to their cellular characteristics and arrangements, as well as the presence of cystic spaces within the tumor bulk. These results are in line with previous research findings [20,22,23,24,30,31,32].

While our study identified several features that can effectively differentiate between different types of PF tumors, not all features are equally informative. Some features may exhibit significant overlap between different tumor types, which can limit their usefulness in certain scenarios. For instance, our study demonstrated that T1_Tumor features did not exhibit any notably distinctive distributions in distinguishing between different tumor types (Figure 3e and Figure 5a–f). However, it could be seen that T1_Ratio is a crucial factor in differentiating PA from other tumor types (Figure 5b,e,f). Additionally, T1CE displays notable distinctive distributions when differentiating all other tumor types from BG (Figure 3f and Figure 5c,d,f).

Based on the pairwise analysis of the dataset, our findings suggest that MB exhibits a more distinct set of MRI features that are strongly correlated with the tumor type. Conversely, PA appears to be more heterogeneous in terms of its MRI features, and the MRI features associated with BG and EP may not be well defined. Furthermore, the positive correlation between both T2 and ADC features may reflect the diverse nature of these tumor types, with different subtypes exhibiting distinct MRI features. The negative correlation observed between DWI and ADC, as well as DWI and T2 features, may reflect differences in tumor cellularity and tissue microstructure. This finding may have important implications for treatment planning, particularly with regard to therapies that target the tumor microenvironment. The findings of this study offer significant insights into the correlations between tumor types and MRI features.

In an ML classification model, a feature’s ability to distinguish between different classes, such as different tumor types, is determined by the degree of separation or overlap between the distributions of the feature values for each class. When the distributions are close together and have significant overlap, the feature is unlikely to provide much discriminative value and will have little impact on the classification decision. Conversely, when the distributions are far apart and have minimal overlap, the feature is more likely to provide discriminative value and will significantly impact the classification decision. Examining the distribution of tumor types across various features can help to identify potential biomarkers that may be useful for diagnostic or prognostic purposes.

Our preliminary analysis in this study agrees with the results obtained from the RF model, thus substantiating its decision-making process. However, there is a possibility of the ML model selecting a non-distinctive feature as the most critical factor, which is irrational. Therefore, it is imperative to provide an explanation for the model’s decisions. To this end, we have proposed a methodology to elucidate the correlation between the KDE analysis and the averaged feature importance of the ML results, as presented in Figure 6, to bring clarity to this association.

The performance of ML models can be significantly influenced by the distribution of samples in the training and testing sets, even if the samples belong to the same class. To ensure more generalizable results, we utilized stratified random sampling to evaluate the dataset across five different distributions. The feature importance was then computed and averaged over the five distributions, as demonstrated in Figure 5 and Figure 6. However, there was still a considerable degree of variability in the results for each distribution (Appendix A). Thus, it is crucial to take into account the sample distribution when assessing ML models and to implement stratified random sampling to ensure robust and generalized outcomes.

We evaluated the performance of eight different ML models and determined that LR is suitable for binary classification tasks. However, in discriminating between all tumor types simultaneously, RF outperformed all other models. To enhance the interpretability of the RF model’s results, we aligned its feature importance values with the KDE predictions. To ensure reproducibility, we used five different random seeds and computed the mean of the resulting outputs. Comprehensive analysis of the ML results side by side revealed that no single model outperformed the others, as demonstrated in Table 2. Furthermore, the results differed depending on the models and data structure. To demonstrate this, we compared the RF and XGB models for PF tumor classification, as shown in Figure 5. This approach provided a clear understanding of how the models’ behavior influenced the results, which was crucial for the PF tumor classification task.

Analyzing patient distributions can provide insights into subtypes with unusual patterns that ML models may not detect, leading to errors in calculating tumor characteristics. These outliers can reveal unique features that improve the reliability and accuracy of ML models for medical diagnosis and treatment. Clustering and explaining ML models with larger labeled datasets could enhance our understanding of the heterogeneity within patient subtypes in future studies.

In clinical practice, tumors are assessed based on location, the effect exerted by the tumor on the surrounding tissue, and tumor behavior, including the tendency to invade surrounding tissues or the presence of cystic components or calcification. Tumors with classic imaging features in pathognomonic locations can be identified even by novice radiologists with ease. However, distinguishing between tumors with similar characteristics and locations requires a more in-depth analysis of the imaging features, as demonstrated in this study. When two tumors exhibit near-similar characteristics and locations, the ability to differentiate them based on imaging features such as those studied in this work becomes important. AI models trained on MRI sequences can assist in diagnosing similar lesions and aid in management planning. The transparent use of ML methods with pre-analysis and proper testing procedures is crucial for reliable, reproducible, and accurate findings. Ultimately, the primary goal of any analysis is to produce explainable and reproducible results that can be verified by other researchers, improving the diagnostic features and patient outcomes in medical research.

There are some limitations that need to be considered in the present study. First, the dataset used for analysis was limited in scope and size. Although it contained a sufficient number of samples to train ML models, the dataset may not have been representative of all possible scenarios, and the results may not generalize well to other datasets. To address this, we employed stratified random sampling to ensure that each tumor subtype was represented proportionally in the training and testing datasets. This approach helped to minimize bias in the model training and increase the generalizability of our findings. Second, our dataset only included four types of pediatric PF tumors, which may not fully represent the diversity of pediatric brain tumors. Third, the study was limited to the analysis of a single feature and pairwise interactions between features. Other important features or higher-order interactions that were not considered in this analysis may exist, and their inclusion may change the outcome. Future studies with larger sample sizes and additional advanced MRI protocols, such as semiquantitative and quantitative perfusion MRI and MR spectroscopy, could provide more insights into the diagnostic and prognostic value of MRI features for pediatric PF tumors. Additionally, further research is needed to investigate the potential of ML models and EDA to improve the reliability of pediatric PF tumor diagnosis and treatment.

## 5. Conclusions

The significance of our study lies in its ability to surpass the constraints of prior research in this field. While previous studies have often focused on only one binary differentiation or incorporated numerous exceptions, leading to reduced transparency, our research stands out by offering a comprehensive analysis of four distinct tumor subtypes within a single source. This paper offers a comprehensive and holistic understanding of the subject matter.

Through our analysis, we have uncovered the effectiveness of specific MRI features, such as ADC and DWI, in accurately distinguishing between tumor types, while also shedding light on the limited impact of features such as T1 and T1CE. The combination of EDA and ML has provided valuable insights into feature distributions and their importance in classification. Additionally, handcrafted features such as T2_Ratio and T1_Ratio enhanced the effectiveness of T2 and T1 features, respectively, in tumor classification. Overall, we identified RF as a suitable model for tumor classification, while LR emerged as the optimal choice for most binary cases.

In our analysis, we focused on MRI features that demonstrated minimal overlap between tumor types within their KDE distributions, as they offered valuable discriminatory information. Our findings have highlighted the potential of specific features, such as ADC_Ratio and ADC_Tumor, in effectively differentiating between tumor types. This effectiveness can be attributed to the distinct cellular characteristics, arrangement, and presence of cystic spaces within the tumor mass.

We have also demonstrated that in situations where patient data are limited, complex systems may not always be necessary to evaluate feature importance; in fact, they could impair both performance and interpretability. By conducting comprehensive analyses using simpler approaches, we can still extract valuable insights into the significance of specific features. This emphasizes the importance of adaptability and resourcefulness in leveraging available data to make informed decisions in clinical settings.

## Figures and Tables

**Figure 1 cancers-15-04015-f001:**
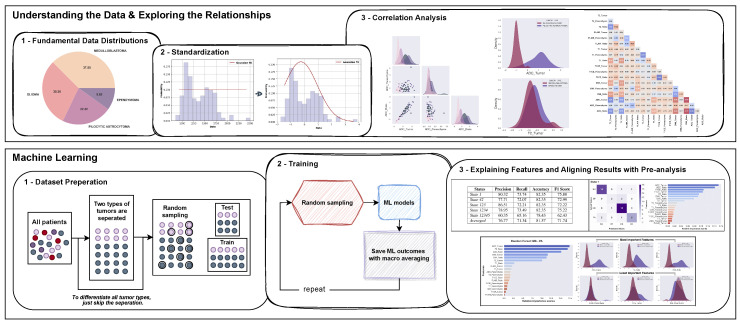
A flowchart depicting the proposed analysis for the classification of pediatric PF tumors: standardization of the dataset, pairwise feature analysis to examine various features of PF tumor types, and aligning interpretations of pre-analysis with ML models’ outcomes.

**Figure 2 cancers-15-04015-f002:**
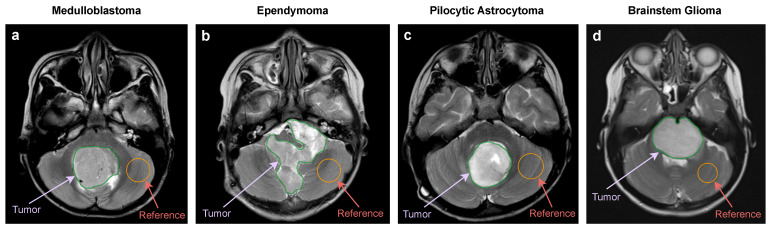
Example of ROI delineation on a T2W MRI. (**a**) MB: 8 years old, boy. (**b**) EP: 3 years old, boy. (**c**) PA: 7 years old, girl. (**d**) BG: 6 years old, girl.

**Figure 3 cancers-15-04015-f003:**
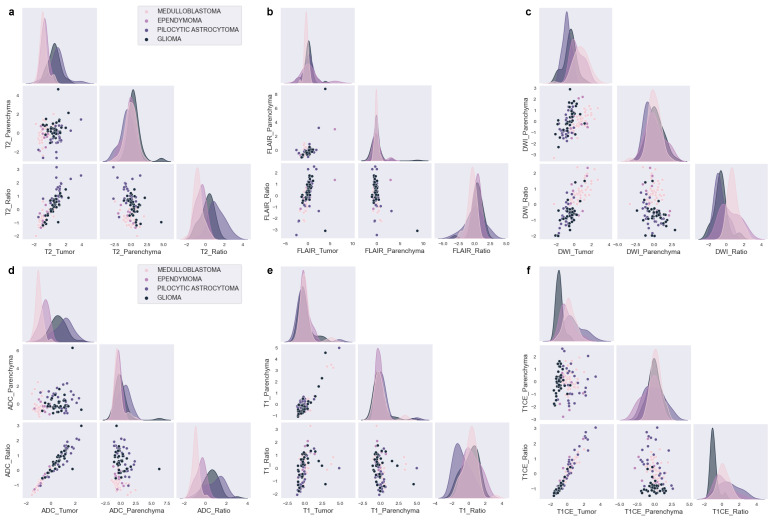
Kernel density estimations with Gaussian distributions of MRI features for PF tumors. (**a**) T2, (**b**) FLAIR, (**c**) DWI, (**d**) ADC, (**e**) T1, and (**f**) T1CE.

**Figure 4 cancers-15-04015-f004:**
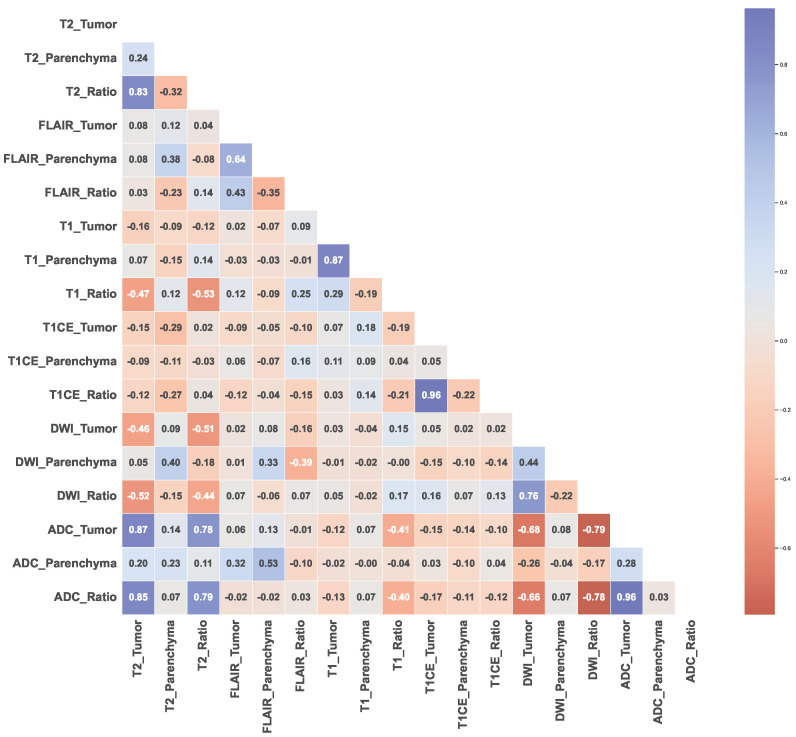
Pearson’s correlation coefficients between each MRI feature.

**Figure 5 cancers-15-04015-f005:**
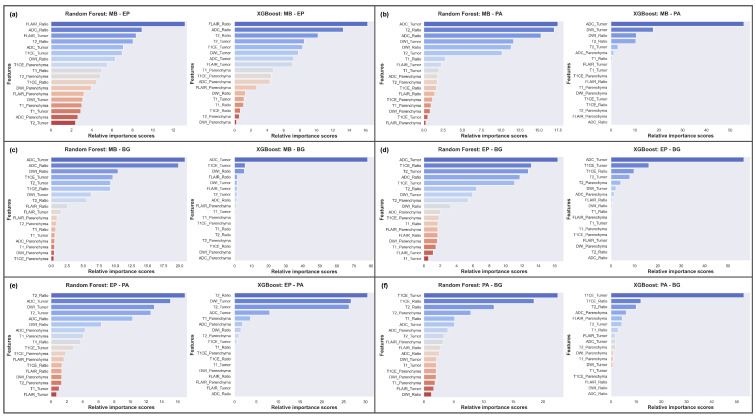
Averaged feature importance scores generated by RF and XGB models for behavior comparison. (**a**) MB-EP; (**b**) MB-PA; (**c**) MB-BG; (**d**) EP-BG; (**e**) EP-PA; (**f**) PA-BG.

**Figure 6 cancers-15-04015-f006:**
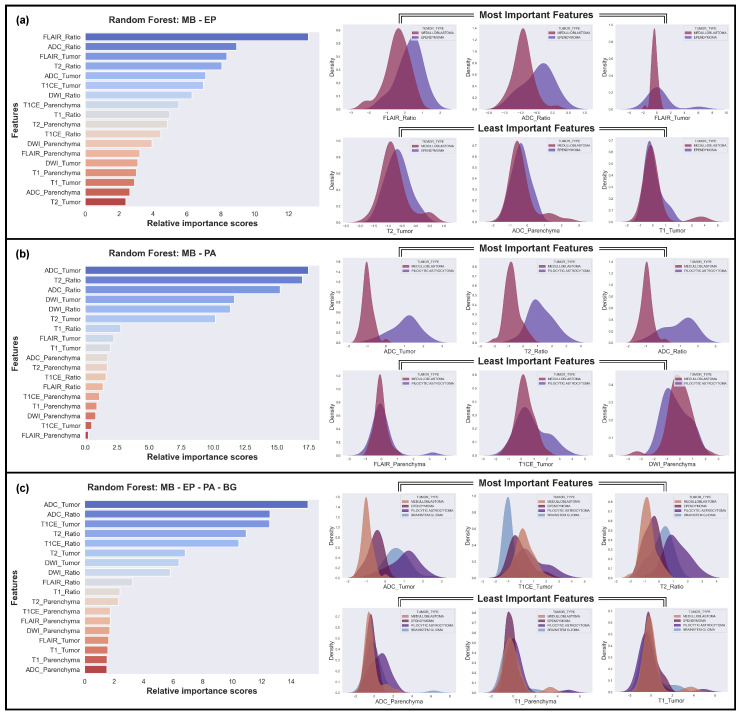
The three most and least effective features for the classifications using the Random Forest (RF) model. (**a**) The hardest case: MB vs. EP; (**b**) The easiest case: MB vs. PA; (**c**) Case of all tumor types.

**Table 1 cancers-15-04015-t001:** Patient demographics.

			Age			Weight		
Tumor Type	Gender	Count	Mean ± Std	Min	Max	Mean ± Std	Min	Max
**Medulloblastoma**	**Girl**	16	7.16 ± 3.74	0.6	13	20.81 ± 8.53	8	35
	**Boy**	26	6.77 ± 3.40	1	13	21.19 ± 8.51	9	40
**Ependymoma**	**Girl**	3	5.67 ± 0.58	5	6	20.33 ± 4.16	17	25
	**Boy**	8	4.00 ± 3.42	1	11	18.38 ± 12.35	9	45
**Pilocytic Astrocytoma**	**Girl**	11	8.18 ± 3.66	3	14	25.18 ± 12.16	9	48
	**Boy**	14	5.79 ± 3.24	1	12	24.07 ± 10.22	10	44
**Brainstem Glioma**	**Girl**	14	6.43 ± 3.69	1	15	22.86 ± 11.81	3	47
	**Boy**	20	6.65 ± 2.85	3	14	24.95 ± 7.82	15	48

**Table 2 cancers-15-04015-t002:** Best test scores for each case from evaluation of 8 different ML models.

	Best Model	Precision	Recall	F1 Score
**MB-EP**	LR	70.65 ± 4.55	68.21 ± 7.86	67.70 ± 6.19
**MB-PA**	LSVM	97.46 ± 1.57	97.28 ± 1.70	97.28 ± 1.52
**MB-BG**	LR	94.94 ± 2.38	94.77 ± 2.50	94.81 ± 2.42
**EP-PA**	LR	95.90 ± 4.17	96.33 ± 4.11	95.80 ± 3.85
**EP-BG**	LR	91.46 ± 7.58	87.50 ± 10.94	87.48 ± 9.73
**PA-BG**	CB	89.95 ± 5.97	88.69 ± 6.86	89.04 ± 6.50
**MB-EP-PA-BG**	RF	76.77 ± 9.78	71.34 ± 3.53	71.74 ± 5.41

## Data Availability

The datasets generated and/or analyzed during the current study are not publicly available due to privacy concerns but are available from the corresponding author upon reasonable request. The source codes, EDA outputs, and entire ML evaluation results of the presented study can be accessed at https://github.com/toygarr/save-the-kid.

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
