# Peer review of "Deciphering Machine Learning Decisions to Distinguish between Posterior Fossa Tumor Types Using MRI Features: What Do the Data Tell Us?"

_cancers, 2023, doi:10.3390/cancers15164015_

Round 1

Reviewer 1 Report

Provide more details on the preprocessing steps applied to the MRI data. Discuss any image registration, normalization, or filtering techniques used to ensure consistency and improve the quality of the input data for the ML models. Elaborate on the specific ML algorithms employed in the study. Describe the architecture and parameters of the models in more depth. This will help readers understand the technical aspects of the models and reproduce the experiments if desired. Clarify the rationale behind choosing kernel density estimation (KDE) with Gaussian distributions for analyzing individual MRI features. Discuss the advantages of this approach over other density estimation techniques and provide references to support its appropriateness for the task. Discuss the methodology for conducting the features' pair relationship analysis. How were the relationships between different features quantified or visualized? Consider providing examples or diagrams to illustrate the approach and facilitate understanding. Explain how the bidirectional validation approach was implemented. Provide a step-by-step description of the process, including any specific algorithms or techniques used. This will help readers follow the methodology and replicate the analysis if desired. Discuss the computational resources utilized for training and evaluating the ML models. Provide information on the hardware specifications, software libraries, and frameworks used. This will help assess the scalability and practicality of the proposed approach. Provide insights into the interpretability techniques used to analyze the ML model behavior. How were the decisions of the models interpreted or visualized? Did you employ techniques such as feature importance analysis, saliency maps, or gradient-based methods? Explain the technical aspects of these interpretability methods. Discuss the statistical significance testing performed, if any, to validate the identified distinguishable MRI features. How were the differences between tumor types quantified and assessed for significance? Provide details on the statistical tests used and the chosen significance threshold. Include a section on potential challenges or limitations related to the ML models used. Discuss issues such as model overfitting, bias, or sensitivity to hyperparameters. Address any steps taken to mitigate these challenges and ensure the reliability of the results. Provide code or pseudo-code snippets for the key algorithms or analysis techniques employed. This will facilitate reproducibility and help other researchers implement and validate the proposed approach in their own studies.

minor edit required

Author Response

Thank you for your feedback. We have thoroughly reviewed your comments and have made the following revisions to the manuscript. Please see the attachment.

Reviewer 2 Report

General Comments: The article addresses a current and interesting topic, focusing on the use of machine learning (ML) for distinguishing between different types of posterior fossa tumors using MRI features. The authors have correctly designed the evaluation methodology and conducted extensive statistical analysis, including the utilization of ML approaches. However, there are several areas that require improvement and clarification to make the presentation of results more accessible and concise. 

Specific Comments and Suggestions: 

  • Presentation of Results: The presentation of results is not suitable in its current form and needs to be revised for clarity and comprehensibility. Instead of presenting all the obtained results in the form of extensive graphs and tables, it is advisable to selectively present significant findings. Comprehensive tables can be included in an appendix for reference, but the presentation of results should be specific to the analysis and the hypotheses under consideration. Consider dividing the results into sections to facilitate analysis and understanding. 

  • Citations: The article lacks recent citations from 2022 and 2023. It is important to incorporate the most up-to-date references to ensure the relevance and accuracy of the work. 

  • Comparison to Existing State-of-the-Art: It is essential to discuss how the proposed analysis differs from previous studies and highlight the novelty and contribution of the current work. What sets this analysis apart from previous approaches? In what aspects does it provide new insights? 

  • Clinical Relevance: The article focuses solely on distinguishing between tumor types without explicitly connecting the proposed classification to diagnosis or prognosis prediction. Is it challenging for radiologists to differentiate between tumor types visually? While the analysis of different modalities and the exploration of the ratio and tumor values are valuable contributions, it would be beneficial to emphasize the practical implications and clinical significance of the proposed classification. 

  • Region of Interest (ROI) Selection: The article lacks information regarding how the ROIs were defined and which specific regions were chosen (e.g., small tumor regions, whole tumor segmentation). It is crucial to clarify the ROI selection process and its impact on the results. What is the variability of ROI values, and how does it affect the outcomes? 

  • Standardization: The equation used for standardization is not adequately described. Instead of the variable "mu" there is the variable "u". Specify the values of "mu" and "std" and clarify what they represent. Is it the mean and standard deviation computed across patients? The equation needs to be described in more detail for better understanding. 

  • Single Analysis and Different Distributions: The use of visual inspection of histograms for the single analysis is mentioned, but it is unclear what is meant by "different distributions" and how the differentiation is established. Could a ROC analysis with global thresholding be performed to quantify this analysis? It would be beneficial to provide a quantitative assessment of this analysis. 

  • Correlation Analysis and Kernel Density Estimation (KDE): The article mentions correlated features identified through KDE analysis. How does this differ from Pearson correlation analysis? Are there any novel findings with KDE analysis in comparison to Pearson correlation? It is necessary to elucidate the added value of KDE analysis in uncovering correlated features and their subsequent exclusion from the analysis. 

  • Conclusion: The conclusion is written too generally and lacks a clear statement regarding the contributions and innovations of the article. It would be beneficial to provide a concise summary of the identified relationships, correlations, and the significance of the features. Additionally, consider including recommendations for disease diagnosis and potential guidelines based on the study's findings. This might even merit a special subsection in the result section. 

  • Reproducibility Analysis: In terms of reproducibility, it is preferable to report the variance of the experiments rather than the complex results. The provided a formula for the average model is redundant. Avoid unnecessary details in the text and focus on quantitative measurements that convey the information effectively. 

  • Metrics and Figures: The rationale for selecting the four metrics used in the study should be discussed and properly interpreted. It is important to provide a comprehensive understanding of the metrics' purpose and what they convey to the reader. Furthermore, Figure 7 appears overly complex, and it would be better to conduct a numerical analysis of model variability. Instead of presenting all the results, consider quantifying them with a few relevant numbers to enhance clarity. 

  • Alternative Expressions: The inclusion of alternative expressions of equations 4-6 is unnecessary since they describe the same concepts and are not explicitly utilized when the functions from the package are already being used. 

  • Random State: A random state is a state seed for a pseudo-random number generator, right? It is used to ensure reproducibility by initializing the random number generator with a specific seed value. Clarify this point in the article to provide a clear understanding to the readers. A verbal description of the results of the individual seeds is not an appropriate presentation of the results, it is better to make an analysis of the variability of the obtained models and present it quantitatively. 

In summary, the article addresses a relevant topic and employs appropriate methodology and statistical analysis. However, it requires substantial revisions to improve the presentation of results, discuss clinical implications, clarify certain aspects, and provide a better summarization and specific analysis. The article may be published after incorporating the submitted suggestions that will enhance the overall quality, clarity and impact of the research article. In particular, it is necessary to change the style of presenting the results, not by presenting all the results, but to select and highlight only the results that are related to the established hypotheses and bring basic and major new insights.

Round 2

Reviewer 1 Report

Accept

Accept